# The essential role of Dnmt1 in *gametogenesis* in the large milkweed bug *Oncopeltus fasciatus*

Joshua T Washington[1], Katelyn R Cavender[1], Ashley U Amukamara[1], Elizabeth C McKinney[1], Robert J Schmitz[2], Patricia J Moore[1]*

[1]Department of Entomology, University of Georgia, Athens, United States;
[2]Department of Genetics, University of Georgia, Athens, United States

**Abstract** Given the importance of DNA methylation in protection of the genome against transposable elements and transcriptional regulation in other taxonomic groups, the diversity in both levels and patterns of DNA methylation in the insects raises questions about its function and evolution. We show that the maintenance DNA methyltransferase, DNMT1, affects meiosis and is essential to fertility in milkweed bugs, *Oncopeltus fasciatus*, while DNA methylation is not required in somatic cells. Our results support the hypothesis that *Dnmt1* is required for the transition of germ cells to gametes in *O. fasciatus* and that this function is conserved in male and female gametogenesis. They further suggest that DNMT1 has a function independent of DNA methylation in germ cells. Our results raise the question as to how a gene that is so critical to fitness across multiple insect species is able to diverge widely across the insect tree of life.

## Introduction

Despite the apparent ubiquity of DNA methylation across the eukaryotic tree of life (*Schmitz et al., 2019*; *Lewis et al., 2020*), in the insects there is considerable variation both in the presence and extent of DNA methylation and even the presence and number of the DNA methyltransferases (*Bewick et al., 2016*; *Lyko, 2018*; *Glastad et al., 2019*). Thus, the functional role of DNA methylation and its associated DNA methyltransferases in the insects is unclear. While some studies have associated levels of methylation with gene expression, most studies in the insects find no functional association (*Bewick et al., 2016*; *Glastad et al., 2019*).

One role emerging from knockdown and silencing studies across several insect species is that *Dnmt1* is required for oogenesis (*Schulz et al., 2018*; *Bewick et al., 2019*; *Glastad et al., 2019*; *Amukamara et al., 2020*; *Gegner et al., 2020*). For example, *Dnmt1* is required for maintenance of DNA methylation following cell division in the milkweed bug, *Oncopeltus fasciatus* (*Bewick et al., 2019*; *Amukamara et al., 2020*), the downregulation of *Dnmt1* using RNAi results in a reduction in methylation and also the cessation of oogenesis. However, it is unclear whether the effect on oogenesis is mediated by the reduction in DNA methylation (*Amukamara et al., 2020*). While the expected reduction in DNA methylation is seen throughout the organism following *Dnmt1* knockdown, the only phenotypic consequence is to the germ cells. Evidence from other species supports a function independent of DNA methylation. *Dnmt1* is also essential to egg production in *T. castaneum* (*Schulz et al., 2018*), a beetle that has no DNA methylation at all (*Zemach et al., 2010*). This suggests that *Dnmt1* can have a function specific to germ cells in insects that is independent of its function in maintaining DNA methylation. This led us to hypothesize that *Dnmt1* plays a role in meiosis in insects (*Amukamara et al., 2020*).

In this study, we tested the hypothesis that *Dnmt1* is essential to gametogenesis in *O. fasciatus*. Furthermore, if *Dnmt1* has a role in gametogenesis, including meiosis, it should be conserved across

*For correspondence:
pjmoore@uga.edu

Competing interests: The authors declare that no competing interests exist.

the sexes. Testing the function of *Dnmt1* in males allows us to capitalize on the well-characterized process of spermatogenesis in *O. fasciatus*, specifically (illustrated in *Figure 1*; *Economopoulos and Gordon, 1971*), and the conserved features of insect spermatogenesis, generally (*Dallai, 2014*). In *O. fasciatus,* as in many insects, there are two points in development where meiosis can occur. The first stage where meiosis will occur is during the larval stages. Testis development and spermatogenesis are initiated during larval development in *O. fasciatus* (*Economopoulos and Gordon, 1971*). During the first three instars the testes consist of seven globular follicles that will develop into the testis tubules (*Schmidt et al., 2002*). Meiosis is initiated in the fourth instar, and by the end of the fourth instar, cysts containing spermatids are present. Differentiation of the spermatids commences in the fifth instar, and males emerge with up to 250,000 spermatids that continue to differentiate during sexual maturation. Thus, we can target the developmental time point at which meiosis occurs. In addition, *O. fasciatus* males can produce gametes throughout their adult lives and therefore meiosis can occur in the adult testis. If *Dnmt1* is required for meiosis and gametogenesis we predicted that it will be required both during larval development and as adult males replenish sperm stores following mating. We therefore compared the testis phenotypes of adults developing from nymphs in which *Dnmt1* expression was downregulated either before or after the major wave of meiosis that occurs in testis development (*Economopoulos and Gordon, 1971*; *Schmidt et al., 2002*; *Ewen-Campen et al., 2013*). We also treated sexually mature adults and examined fertility in males following sperm depletion, testing for the ability of knockdown males to replenish sperm stores following multiple matings. Our results demonstrated that expression of *Dnmt1* is required for the development of sperm both during larval and adult spermatogenesis and that the impact of *Dnmt1* knockdown was greatest if it occurred prior to the onset of meiotic divisions in the developing testes. These results establish that *Dnmt1* plays a critical conserved function across the sexes during gametogenesis in *O. fasciatus* and that it is required for germ cell development.

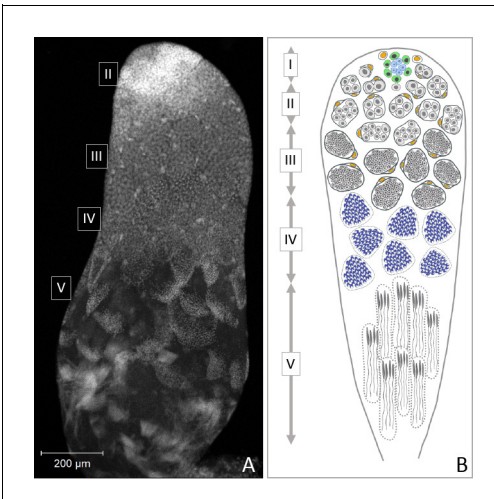

**Figure 1.** Progression of spermatogenesis in *O. fasciatus.* In *O. fasciatus* spermatogenesis progresses from the tip of the testis tubule. (**A**) DAPI-stained testis tubule of stock male. (**B**) Diagram of the stages of spermatogenesis. At the apical tip (region I) of each of the seven testis tubules there is a 'rosette' of niche cells (blue) surrounded by the germline stem cells (**B**, green; *Schmidt et al., 2002*). The rosette is not typically visible in confocal images of testis tubules and markers for the rosette have not been identified yet. As spermatogonia (**B**, light gray) arise from division of the germline stem cells, they are enclosed by cyst cells (**B**, yellow). In region II, spermatogonia undergo mitotic transit amplification divisions to form spermatocysts containing 64 spermatogonia (*Economopoulos and Gordon, 1971*; *Ewen-Campen et al., 2013*). Spermatocytes (**B**, dark gray) in region III divide meiotically. *Oncopeltus fasciatus* undergoes inverted meiosis (*Viera et al., 2009*). Primary spermatocytes undergo the first meiotic division to produce diploid secondary spermatocytes. The meiotic division of the secondary spermatocytes produces the haploid spermatids (**B**, dark blue) in region IV that then differentiate into spermatozoa in the region V at the terminal end of the testis tubule.

## Results

### The pattern of *Dnmt1* expression during testis development and across tissues suggests a role in spermatogenesis

We measured *Dnmt1* expression during larval development and sexual maturation. Our prediction was that if *Dnmt1* is required during gametogenesis, particularly meiosis, then its expression should mirror those of two genes known to be involved in germ cell development, *Boule* and *Vasa* (*Shah et al., 2010*; *Yajima and Wessel, 2011*). Thus, we predicted that *Dnmt1* expression would be highest during the stages when gametogenesis is occurring. As predicted, testis-specific expression of *Dnmt1* is highest during the fourth and fifth instars of development (*Figure 2A*; ANOVA, F = 27.426, d.f. = 3,

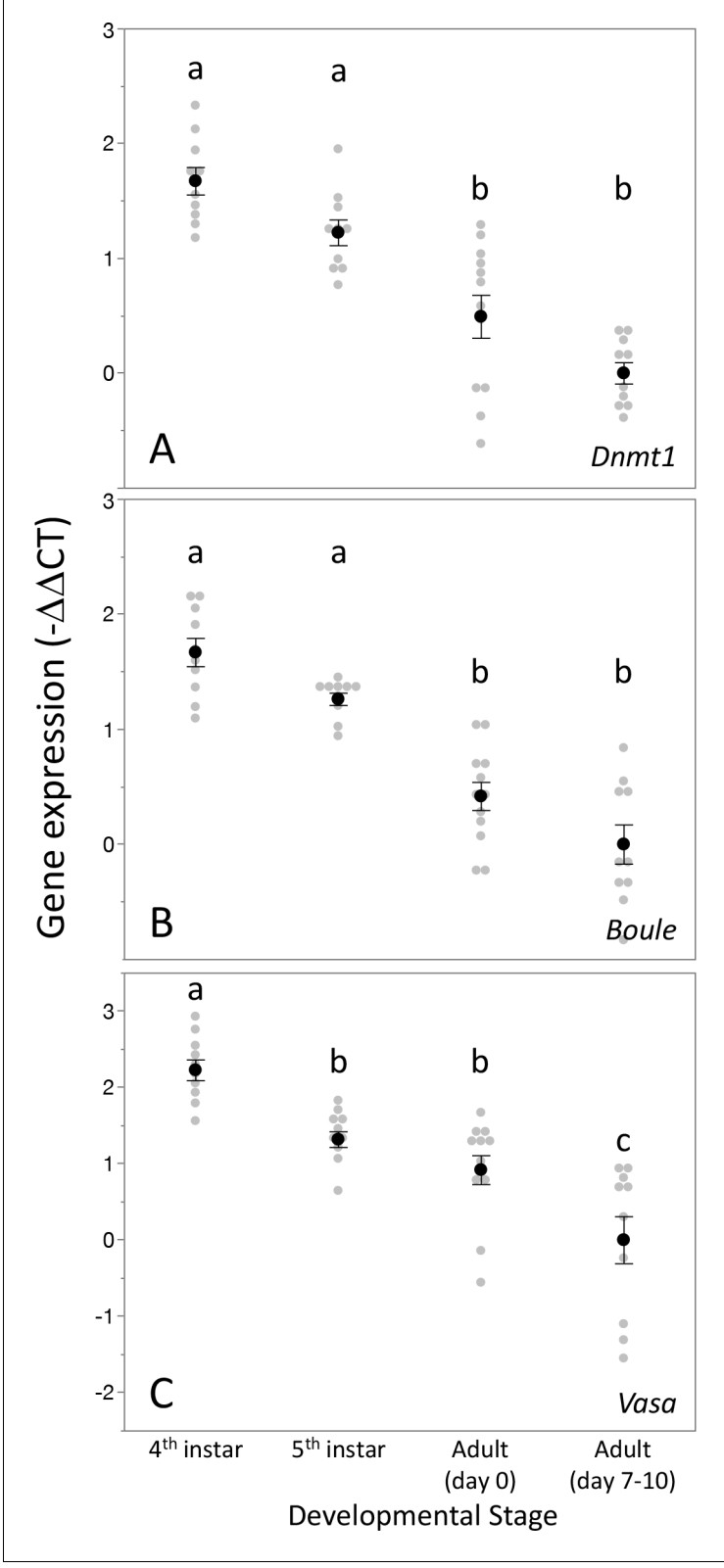

**Figure 2.** Expression pattern of *Dnmt1* within the developing testes mirrored that the two gametogenesis genes *Boule* and *Vasa. Dnmt1* expression (**A**) was highest in the larval stages where spermatogenesis is initiated compared to the testes of newly emerged (day 0) and virgin sexually mature (days 7–10) adults. This is similar to the expression patterns of *Boule* (**B**), in which expression was higher in the fourth and fifth larval stages during
*Figure 2 continued on next page*

*Figure 2 continued*

which the transition to meiosis occurs compared to newly emerged and virgin sexually mature adults. Expression of *Vasa* (C) demonstrated a similar pattern, with the highest expression in the fourth instar stage and lowest in sexually mature virgin males. Black dots and bars represent mean and SE. Gray dots represent data points for each individual tested. Lowercase letters designate significant differences (p<0.05) among pairwise comparisons using post hoc Tukey–Kramer HSD test.

The online version of this article includes the following source data and figure supplement(s) for figure 2:

**Source data 1.** Expression of Dnmt1 in testes across developmental stages.
**Figure supplement 1.** Expression of *Dnmt1* is highest in reproductive tissue.
**Figure supplement 1—source data 1.** Expression of Dnmt1 across tissues in males.

38, p<0.001), stages when meiosis and spermatogenesis are most active (*Economopoulos and Gordon, 1971*). The expression is lowest in virgin adult males, both at adult emergence and at sexual maturation. This expression pattern mirrored that observed for *Boule*, a gene with a well-characterized role in meiosis (*Figure 2B*; ANOVA, F = 36.346, d.f. = 3, 38, p<0.001). *Dnmt1* expression pattern was also similar to *Vasa*, a highly conserved marker of the germline (*Figure 2C*; ANOVA, F = 20.444, d.f. = 3, 38, p<0.001). While *Dnmt1* is expressed across all adult tissues, as would be expected given its role in maintenance of DNA methylation, expression is highest in testes compared to gut, head, and muscle tissue (*Figure 2—figure supplement 1*; ANOVA, F = 67.311, d.f. = 3, 36, p<0.001), also suggesting a special role during gametogenesis.

## *Dnmt1* knockdown reduced levels of DNA methylation in the testis genome

To confirm the effectiveness of our RNAi treatment, we measured expression of *Dnmt1* in a subset of ds-RED and ds-*Dnmt1* injected males (*Figure 3*). Quantitative PCR demonstrated reduced levels of *Dnmt1* mRNA in ds-*Dnmt1* injected males compared to ds-RED injected males in both third instar (ANOVA; F = 83.126, d.f. = 1, 31, p<0.001) and fifth instar (ANOVA; F = 49.793, d.f. = 1, 22, p<0.001) treated males 6–10 days following injection with dsRNA. Treatment of males with ds-*Dnmt1* had the predicted effect on *Dnmt1* function in maintenance of methylation. We performed whole genome bisulfite sequencing to evaluate the impact of the ds-*Dnmt1* on DNA methylation genome wide. The reduction in expression of *Dnmt1* in the RNAi individuals had the expected phenotypic effect on DNA methylation with a reduction of genome methylation in the testes for ds-*Dnmt1* treated males, but not control males (*Figure 4*). Genomic DNA from the testes of control treated males had approximately 12.5% CpG methylation regardless of stage they were treated. Knockdown of *Dnmt1* at the earlier stage of development led to a greater percentage reduction of methylation. Treatment with ds-*Dnmt1* at the fifth instar reduced the percent CpG methylation from around 12.5% to around 5%. Treatment with ds-*Dnmt1* at the third instar reduced the level of methylation even further, to around 2%, as predicted given the greater numbers of cell divisions that were expected between treatment and sampling between these two treatments.

## *Dnmt1* knockdown affected testis size and structure

Knockdown of *Dnmt1* prior to meiosis, in the third instar stage, had a significant effect on testis size in virgin males, while knockdown of *Dnmt1* during the developmental stage after which meiosis is initiated, the fifth instar stage, had no effect on testis size (*Figure 5*; ANOVA, F = 20.360, d.f. = 4, 58, p<0.001) compared to either uninjected controls or ds-RED treated males.

Knockdown of *Dnmt1* prior to meiosis affected testis tubules to a greater extent than knockdown following the initiation of meiosis during larval testis development (*Figure 6*). Control males injected in the third or fifth instar stage of development showed the expected structure of the testis tubule (*Figure 6A and C*). At the anterior end of the testis tubule, spermatogonia and primary spermatocytes (testis region II) had the characteristic nuclear structure and there was clear evidence of mitotic division within the spermatogonia (*Figure 6C*, arrowhead), using α-phosphohistone H3 (pHH3) to stain for chromosome condensation in preparation for mitosis and meiosis. We also observed α-pHH3 staining in spermatocysts (testis region III) at the border between primary and secondary spermatocytes in the control testis tubules in both third and fifth instar treated males. Following this

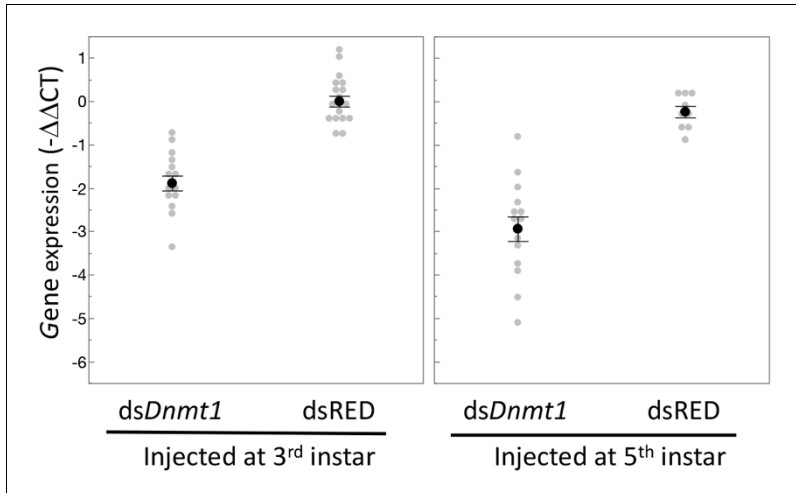

**Figure 3.** Expression of *Dnmt1* was reduced in the testes of adult males treated with ds-*Dnmt1* at both stages of development. Relative gene expression is standardized to expression levels in control treatments. Black dots and bars represent mean and SE. Gray dots represent data points for each individual tested.

The online version of this article includes the following source data for figure 3:

**Source data 1.** Dnmt1 expression is knocked down following ds-Dnmt1 treatment.

band of relatively synchronous meiotic activity, the posterior testis tubule showed spermatids and developing spermatozoa as they matured. We confirmed the use of this band of α-pHH3 stained spermatocysts as a landmark of meiosis using knockdown of *Boule* (*Figure 6—figure supplement 1*).

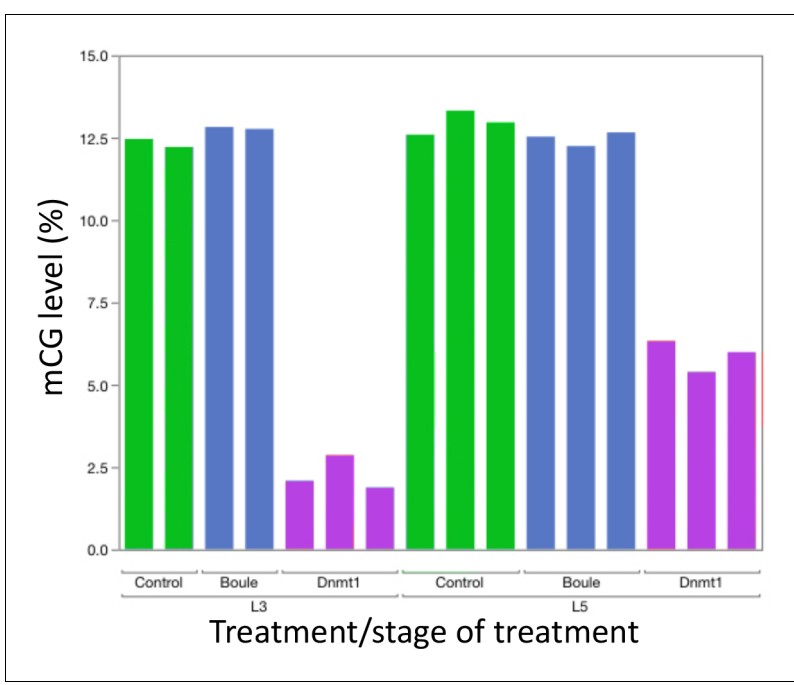

**Figure 4.** Knockdown of *Dnmt1* (magenta bars), but not *Boule* (blue bars), reduced DNA methylation compared to controls (green bars). Each bar represents a single individual. Across all individuals, the percent methyl CpG is reduced in the DNA isolated from the adult testis of males treated with *ds-Dnmt1*. Earlier injection reduces the percent methylation to a greater extent, consistent with more rounds of cell division within the testes between injection and dissection.

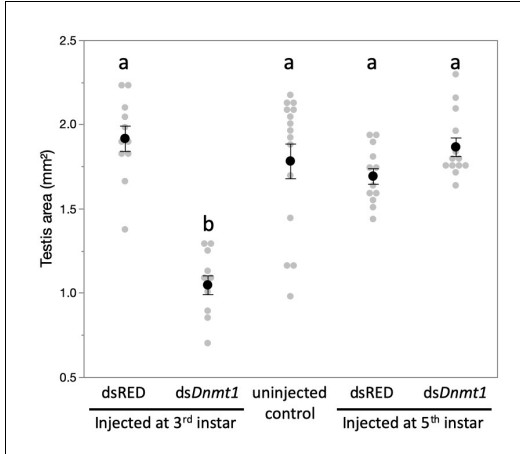

**Figure 5.** Downregulation of *Dnmt1* during the third instar stage of development significantly reduced the size of the testis in sexually mature males. There was no effect of downregulation of *Dnmt1* on testis size when treatment occurred during the fifth instar stage of development after meiosis has been initiated. Black dots and bars represent mean and SE. Gray dots represent data points for each individual tested.

The online version of this article includes the following source data for figure 5:

**Source data 1.** Testis area from adult males treated with ds-Dnmt1 at L3 or L5 stage of development.

The structure of the testis tubules of males treated with ds-*Dnmt1* in the third instar was highly disrupted and both the anterior and posterior testis tubule was affected (*Figure 6B*). There were fewer spermatocysts in both the region of spermatogonia and primary spermatocytes. While there were occasional α-pHH3 positive nuclei, these were not well organized into spermatocysts and were spread throughout the testis tubule rather than organized at the junction between primary and secondary spermatocytes. There was evidence of mitotic activity in the spermatogonia (*Figure 6B* arrowhead), although these were less frequent in ds-*Dnmt1* treated males than control males. The testis tubules of males treated with ds-*Dnmt1* in the fifth instar stage of development had a structure much more similar to control males (*Figure 6D*). Mitotic activity was apparent in the spermatogonia (*Figure 6D* arrowhead) and most testis tubules had evidence of mature sperm and α-pHH3 stained spermatocysts at the junction between primary and secondary spermatocytes (*Figure 6D* arrow). Unorganized spermatocysts below this junction were frequently observed, however, and ds-*Dnmt1* males treated at the fifth instar had variable phenotypes posterior to the primary spermatocytes, presumably depending on when treatment occurred following the wave of meiosis along the testis tubule axis.

## *Dnmt1* knockdown in adult males prevented replenishment of sperm stores

Downregulating *Dnmt1* expression in adult males resulted in a loss of fecundity over time. The third and final females mated to ds-*Dnmt1* treated males ran out of sperm to fertilize eggs more rapidly than those mated to control males. Clutches of eggs laid by females mated to ds-*Dnmt1* treated males were not fertilized and failed to hatch at an earlier collection day than those laid by females mated to control males (*Figure 7*; Wilcoxon $\chi^2$ = 13.978, d. f. = 1, p<0.001).

The loss of fertility in males was associated with smaller testis size. Males treated with ds-*Dnmt1* at 7 days post-emergence and then allowed to mate for 3 weeks prior to dissection had statistically significantly smaller testis area than the testes of control males after the same mating treatment (ANOVA, F = 29.084, d.f. = 1, 51, p<0.001; *Figure 7—figure supplement 1*). The smaller testis area of ds-*Dnmt1* treated adult males was associated with a breakdown in testis tubule structure and the loss of sperm from the testis tubules (*Figure 8*). In control males, 3 weeks post injection the regions of spermatogenesis were recognizable. Small spermatocysts with spermatogonia undergoing mitotic divisions were observable in testis region II at the anterior tip of the testis tubule (*Figure 8A and B*). Posterior to the spermatogonia were the primary spermatocytes with more diffuse nuclear structure. We often observed the band of α-pHH3 positive spermatocysts that indicated the first meiotic division to form the secondary spermatocytes. At the most posterior end of the testis tubule the spermatids developed into spermatozoa. Three weeks after *Dnmt1* knockdown testis tubule structure is significantly disrupted. There appeared to be fewer spermatogonia and those that remained had a more condensed nuclear structure than in the testis tubules of control males (*Figure 8C and D*). The most noticeable effect was seen in testis region III containing the spermatocytes. There were few primary spermatocytes in the testis tubule and α-pHH3 positively stained spermatocysts were rarely observed, indicating few spermatocysts undergoing meiosis.

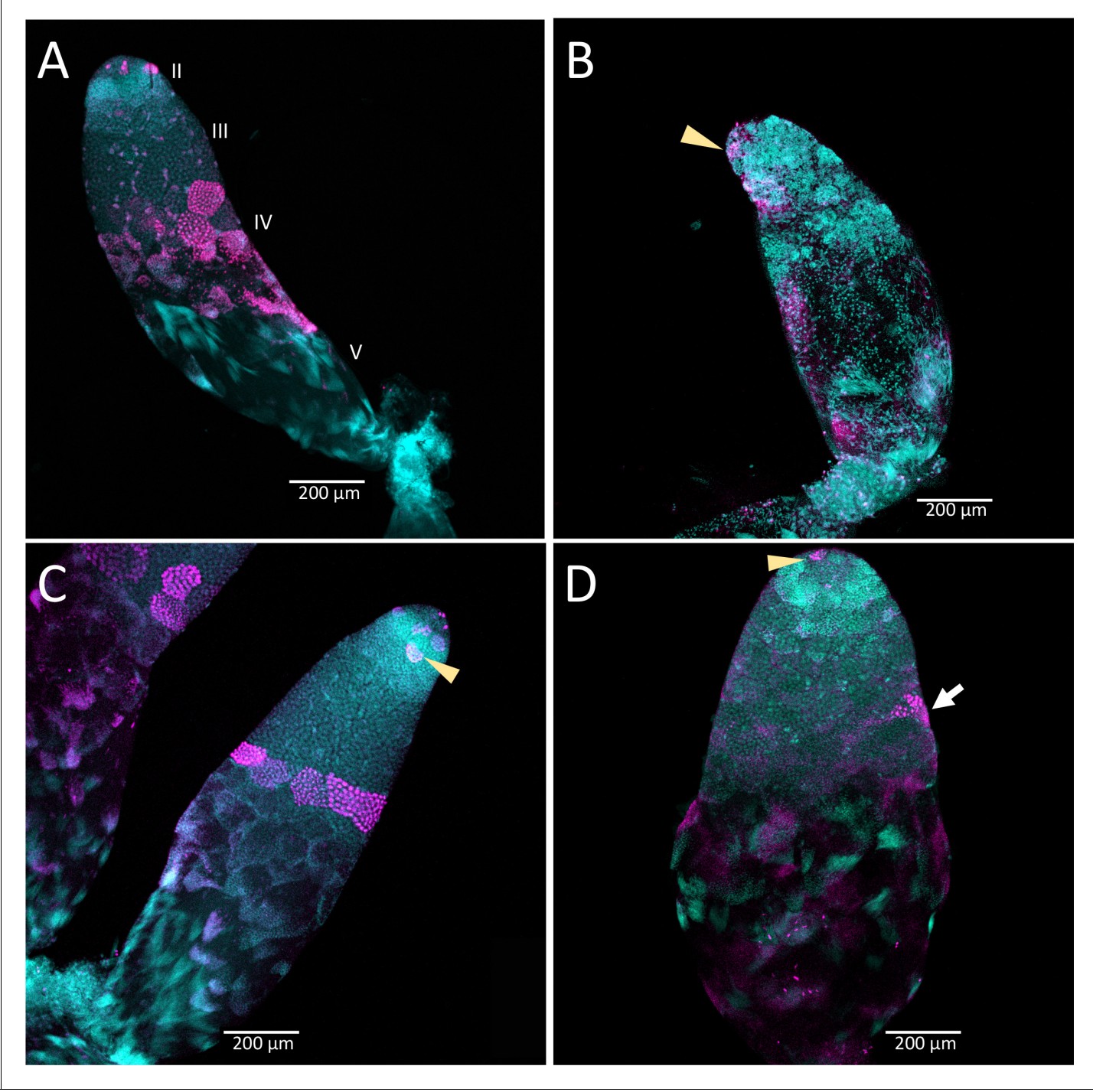

**Figure 6.** Timing of treatment during development determined the effect on testis structure of adults that develop from *Dnmt1* (Panels **B** and **D**) knockdown males. In control testis tubules from males injected with ds-RED (**A and C**), spermatogonia divided mitotically in region II to form spermatocysts as is observed in untreated males. Mitotic spermatogonia are labeled with anti-phosphohistone three antibodies in region II of the testis tubule (**C**, arrowhead). Meiotic divisions also occurred as in untreated males in ds-RED injected males. In our ds-RED control samples, a band of synchronously dividing spermatocysts, identified by positive anti-phosphohistone H3 antibody staining, was present at the interface between primary and secondary spermatocytes in testis region III (**A and C**). In testes from males with *Dnmt1* knockdown in the 3$^{rd}$ instar (**B**), the anterior tip of the testis tubule, region II, showed evidence of mitotic activity (arrowhead). However, there were fewer spermatocytes present and the spermatocysts in this region were disorganized, and there was little evidence of the band of positive anti-phosphohistone H3 stained meiotic spermatocytes in region III. The testis tubule structure from males treated with ds-*Dnmt1* following meiosis at the fifth instar stage of development (**D**) was more similar to the controls than those treated at the third instar stage of development. In ds-*Dnmt1* males treated at the fifth instar stage, there were positive anti-phosphohistone

*Figure 6 continued on next page*

*Figure 6 continued*

H3 stained spermatogonia in region II (arrowhead) and spermatocytes in region III (arrow). There were differences between males treated as fifth instars with ds-RED males and ds-*Dnmt1*, however, including spermatocysts containing cells with highly condensed nuclei that were not present in the controls and fewer spermatocysts containing spermatids or spermatozoa. All images taken at 10× magnification.

The online version of this article includes the following figure supplement(s) for figure 6:

**Figure supplement 1.** *Boule* knockdown with RNAi showed the expected phenotype in males.

## Discussion

The function of DNA methylation and the DNA methylation enzymes across the insect tree of life has been widely debated, particularly in light of the extreme variation in the presence of this chromatin modification and the diversification of the enzymatic toolkit required to methylate DNA de novo and maintain DNA methylation patterns (*Bewick et al., 2016*; *Lyko, 2018*; *Lewis et al., 2020*). This evolutionary pattern is particularly intriguing given that in two species on divergent branches of the insect phylogeny the maintenance methyltransferase DNMT1 is essential to gametogenesis, and thus is tightly tied to fitness, even in a species where DNA methylation itself is absent (*Schulz et al., 2018*). While functional studies have not been done in other species, a summary of expression studies on the DNA methyltransferases across a diversity of insects suggest that these enzymes could be involved in gametogenesis across a number of species, ranging from fire ants, *Solenopsis invicta* (*Kay et al., 2018*), to jewel wasps, *Nasonia vitripennis* (*Zwier et al., 2012*), to migratory locusts, *Locusta migratoria* (*Robinson et al., 2016*), and the brown plant hopper, *Nilaparvata lugens* (*Zhang et al., 2015*). Here we showed that *Dnmt1* expression affects germ cell development. The effect of downregulating *Dnmt1* in males, as in females (*Amukamara et al., 2020*), was specific to the germ cells. The downregulation of *Dnmt1* did result in reduction of DNA methylation, but there were no obvious morphological impacts to the RNAi-treated individuals. Our results show that the function of *Dnmt1* in germ cells is conserved across the sexes within a species, and other results

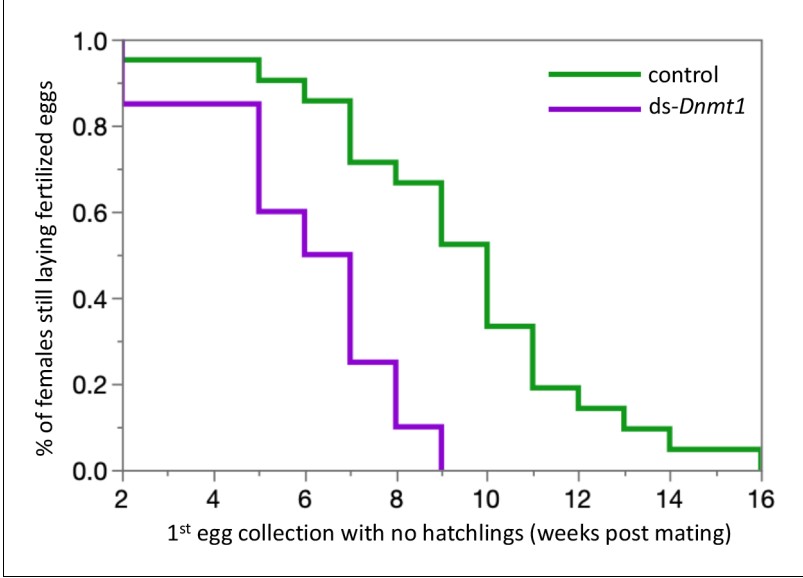

**Figure 7.** Females mated to control males lay fertilized eggs longer than females mated to *Dnmt1* knockdown males. Eggs were collected twice per week and eggs allowed to develop to hatching. Eggs that did not hatch showed no sign of development, indicating that they had not been fertilized. Both treatments demonstrate a decrease in proportion of eggs that hatch over time, but the ds-*Dnmt1* treatment group shows a faster decrease. The online version of this article includes the following source data and figure supplement(s) for figure 7:

**Source data 1.** Fecundity data for the mates of Dnmt1 and control males.

**Figure supplement 1.** Downregulation of *Dnmt1* in sexually mature males resulted in a loss of testis area following multiple matings.

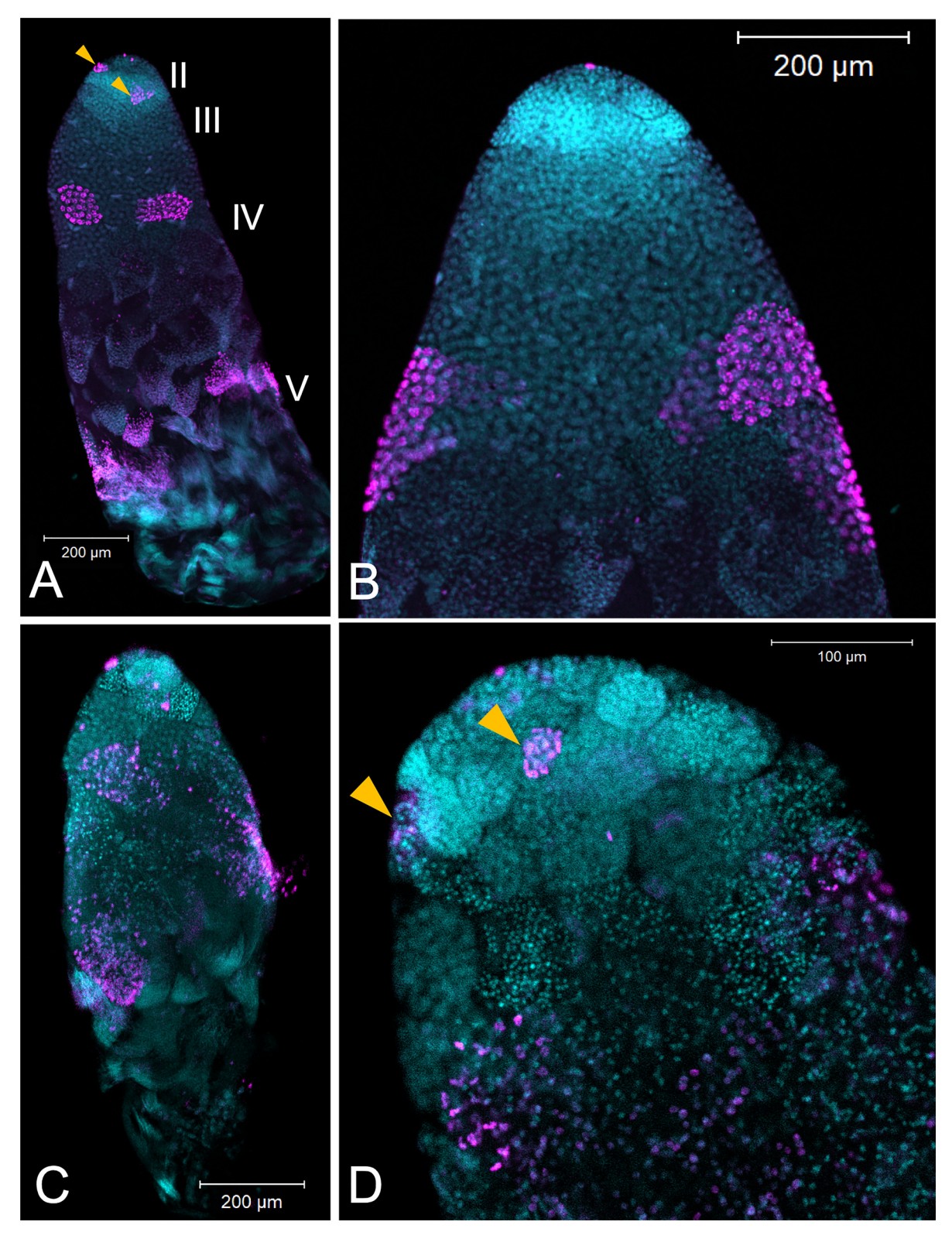

**Figure 8.** Testis structure breaks down in *Dnmt1* knockdown males treated as adults after having sperm replenishment induced by mating activity. The regions of spermatogenesis were apparent in mated males following 3 weeks of mating activity in control testis tubules (**A and B**) and the evidence of both mitotic division in spermatogonia in region II (**B**, arrowhead) and the band of meiotic divisions in region III was clear from anti-phosphohistone H3 staining. In *Dnmt1* knockdown males at low magnification (**C**), the anterior tip, region II, of the testis tubule looked relatively normal. However, region III

*Figure 8 continued on next page*

*Figure 8 continued*

containing both the primary and secondary spermatocytes was disorganized. Spermatocyst structure was broken down, and the nuclei of the primary and secondary spermatocytes had lost their characteristic structure (*Ewen-Campen et al., 2013*). Finally, there were fewer mature spermatids in the posterior end of the testis tubule. At higher magnification (**D**), it was apparent that nuclear structure in the anterior tip was also affected by the knockdown, both in the spermatogonia and spermatocytes. Spermatogonia nuclei in the *Dnmt1* knockdown testis tubules (**D**) were more condensed than in the control testis tubules (**B**), although they still seemed to be organized into spermatocysts. Spermatocyte nuclei, however, were fewer in number than in controls, did not have their characteristic shape (*Ewen-Campen et al., 2013*; *Economopoulos and Gordon, 1971*), and were not organized in spermatocytes. A and C:10× magnification, B and D: 20× magnification.

document that *Dnmt1* expression is required for oogenesis among species. This raises a conundrum when we look across the insect tree of life. How can a gene required for such a fundamental fitness activity in examples from such a diverse group of insects also be so evolutionarily labile? This suggests that the function *Dnmt1* in germ cell development is easily lost or replaced.

## *Dnmt1* knockdown causes inhibition of spermatogenesis consistent with a role in meiosis

If our hypothesis on the role of *Dnmt1* in meiosis is correct, then spermatogenesis should be affected by *Dnmt1* expression as well as oogenesis, as it is a shared process across gametogenesis during sexual reproduction. In the testes *O. fasciatus* males, we observed both spermatogonia and primary spermatocytes that stain positively for phosphorylation of the serine 10 residue of histone protein H3. While the pattern of histone H3 phosphorylation across meiosis has not been specifically studied in *O. fasciatus*, phosphorylation of the serine 10 is typically associated with chromosome condensation during meiosis (*Hans and Dimitrov, 2001*) and mitosis (*Prigent and Dimitrov, 2003*). Moreover, the pattern of α-pHH3 staining along the axis of the testis tubule is consistent with this histone modification occurring in both mitosis and meiosis. We observed α-pHH3 positive cells both within spermatocysts containing the mitotically dividing spermatogonia and also at the boundary between primary and secondary spermatocytes. We have confirmed this landmark using *Boule* knockdown males. In Drosophila, *Boule* is required for progression of spermatocytes through meiosis. Germ cells lacking functional *Boule* become arrested at meiotic prophase (*Eberhart et al., 1996*). In the testes of *O. fasciatus* males treated with ds-*Boule*, the affected cells were all posterior to the band of synchronous α-pHH3 positive spermatocytes. Thus, α-pHH3 staining provided a landmark for entry into meiosis.

We did not see the band of α-pHH3 positive spermatocysts at the border between primary and secondary spermatocytes in the *Dnmt1* knockdown males treated at the third instar stage of development. This could be interpreted as the primary spermatocytes have not initiated chromosome condensation. However, we did observe individual α-pHH3 positive nuclei in the posterior testis tubule. So there could have been arrest after chromatin condensation with a breakdown in spermatocyst structure. Interestingly, the testis phenotype of our *Dnmt1* knockdown males resembled the phenotypic effect of *Vasa* knockdown reported by *Ewen-Campen et al., 2013*. *Vasa* RNAi results in defects in cyst integrity and *Ewen-Campen et al., 2013* propose that *Vasa* plays a specific role in the onset or synchrony of meiosis. *Vasa* has also been proposed to be required for the correct progression through meiosis in mice (*Tanaka et al., 2000*) and humans (*Medrano et al., 2012*). Thus, the phenotypic similarity between *Dnmt1* knockdown and *Vasa* knockdown testes in *O. faciatus* supports the hypothesis that *Dnmt1* may be required for the successful initiation and completion of meiosis in spermatocytes. More work remains to determine exactly what stage of sperm development is impacted by the decrease in *Dnmt1* expression.

The phenotype of knockdown of *Dnmt1* is exacerbated when knockdown occurs prior to the stage of testis development when meiosis occurs. This result is consistent with what was observed in female *O. fasciatus* (*Amukamara et al., 2020*) in which knockdown of *Dnmt1* expression prior to the stage of oocyte development inhibited all oocyte formation. One interpretation of this result is that *Dnmt1* is required for successful progression through meiosis. However, it is clear that reducing *Dnmt1* expression had an impact beyond simply reducing the ability to enter or complete meiosis. A block to progression through meiosis, as demonstrated by the *Boule* knockdown, spermatogonia and primary spermatocytes would continue to be born but remain viable after a failure to complete meiosis. In the *Dnmt1* knockdowns, however, there were fewer germ cells of any type, and those

that remained in the testis tubule often appeared to have abnormal and condensed nuclei. It remains to be determined if *Dnmt1* is required for meiosis or if the association with meiosis is a correlation; *Dnmt1* could be required for viability of germ cells and act at the stage of development at which they would be undergoing meiosis.

Previous studies on the function of *Dnmt1* in insect spermatogenesis have not documented any effect on male fertility. Knockdown of *Dnmt1* in the red flour beetle, *T. castaneum* (*Schulz et al., 2018*) had no effect on the fecundity of female mating partners in the first 9 days post mating and the authors conclude that *Dnmt1* is not required for spermatogenesis in *T. castaneum*. We suspect that the function of *Dnmt1* in spermatogenesis is not unique to *O. fasciatus*, given the expression patterns of *Dnmt1* in other insects (*Robinson et al., 2016*; *Kay et al., 2018*). In the *T. castaneum* study, the authors did not examine testis structure or design a mating strategy that would exhaust the sperm stores of the males and so may have missed a fertility effect. In holometabolous insects, including beetles, spermatocytes are typically formed by the end of larval development (*Economopoulos and Gordon, 1971*), thus males treated as pupae would still emerge with large numbers of sperm in their testes. Here we showed that male *O. fasciatus* treated with ds-*Dnmt1* RNA as sexually mature adults become sperm limited more rapidly than control males. There is a potential alternative explanation that *Dnmt1* knockdown males could continue to produce sperm but that sperm was of low quality and unable to support development. Given that the testes of ds-*Dnmt1* treated males were significantly smaller than control males and had altered testis tubule structure, which indicated that spermatogenesis was significantly impacted, we concluded that *Dnmt1* knockdown males are unable to replenish sperm supplies. The functional role of *Dnmt1* in testes across other species remains to be tested.

## DNA methylation and germ cell development

The knockdown of *Dnmt1* resulted in a reduction of DNA methylation within the testes, as predicted. The effect was greater in the nymphs that were treated in an earlier stage of development, as would be expected given the greater number of cell divisions that would occur between treatment and collection of the testes between the third and fifth instar treatments. The more extreme phenotype in the third instar treated males could be explained by the greater reduction in methylated CpG. However, the percent methylation seen in the testes of the fifth instar ds-*Dnmt1* treated males was similar to that seen in females treated at the fourth instar stage of development (*Amukamara et al., 2020*), greater than a twofold reduction in the percent methylated CpG residues. In *Dnmt1* knockdown females, there was a compete loss of oocyte production when DNA methylation was reduced to this level while in males the phenotype of the fifth instar *Dnmt1* knockdowns was close to normal. This disconnect between reduction of DNA methylation and phenotypic effects mirrors what has been seen in the beetle, *T. castaneum* (*Schulz et al., 2018*). In this species, the reduction of DNA methylation is an evolved difference rather than an experimental effect, but the results are similar to what we have observed in *O. fasciatus*; DNA methylation is not required for function of somatic cells, but downregulation of *Dnmt1* expression leads to specific germ cell effects. The lack of correlation between extent of methylation and function in germ cells strengthens our hypothesis that there may be a pleiotropic function for *Dnmt1* in germ cells that acts independently of DNA methylation levels. Alternatively, DNA methylation may have a specific role in the germ cells that it does not play in somatic cells. Perhaps DNA methylation is required for proper gene expression in germ cells or it may be required for transposon silencing only in germ cells. However, demethylation of ovarian tissue in *O. fasciatus* showed little relationship between methylation levels and gene expression, with few differentially transcribed genes (*Bewick et al., 2019*). Another possibility is that DNA methylation state may be a prerequisite for successful completion of meiosis and not mitosis. For example, *O. fasciatus* has holocentric chromosomes and undergoes inverted meiosis (*Viera et al., 2009*). During mitosis chromosomes have a kinetocore that extends across the majority of the chromosome, but in meiosis the kinetochore plate is missing and microtubules extend into the chromosome (*Comings and Okada, 1972*). One could imagine that DNA methylation might be required for this interaction. However, the complete lack of methylation but a functional role of *Dnmt1* in *T. casteneum* oogenesis alongside our results argues for a pleiotropic function for *Dnmt1* in gametogenesis that is independent of its role in DNA methylation.

## Conclusion

*Dnmt1* expression is required for germ cell development in both male and female *O. fasciatus*. The block to germ cell development in *Dnmt1* knockdowns appeared to be associated with meiosis, although it was not a simple block to progression through meiosis as germ cells are lost from the testes. Thus, *Dnmt1* may be required for germ cell viability. It still remains to be determined if the block in gametogenesis depends on DNA methylation or an alternative function of *Dnmt1*, as suggested by the requirement for *Dnmt1* during gametogenesis in an insect with a non-methylated genome (*Schulz et al., 2018*). Whatever the specific function of *Dnmt1* in gametogenesis, the requirement for this enzyme in such a critical fitness function as the production of gametes in representatives of different groups of insects raises important questions as to how this enzyme and the entire methylation toolkit has evolved across the insect tree of life (*Lyko, 2018*; *Provataris et al., 2018*). Functional analysis of *Dnmt1* in the insect groups where it is found, and study of how these functions are replaced in the species where it is no longer found, will be essential for understanding the evolution of this important base modification.

# Materials and methods

## Key resources table

| Reagent type (species) or resource | Designation | Source or reference | Identifiers | Additional information |
|---|---|---|---|---|
| Strain, strain background | *Oncopeltus fasciatus* | Carolina Biologicals | Item # 143810 NCBI Taxon:7535 | Cultured in Moore lab for 5 years |
| Antibody | Anti-phospho-Histone H3 (Ser10) (Rabbit polyclonal) | Millipore | Cat # 06–570 RRID:AB_310177 | IF (1:1000) |
| Commercial assay or kit | Qiagen RNA easy kit with Qiazol | Qiagen | Cat # 74104 | |
| Commercial assay or kit | LightCycler 480 SYBR Green I Master | Roche | Product No. 04707516001 | |
| Commercial assay or kit | MEGAscript T7 Transcription kit | ThermoFisher Scientific | AMB13345 | |
| Sequence-based reagent | ds-*Dnmt1* RNAi primer set for transcription template | *Amukamara et al., 2020* | PCR primers | Sense: TGATGCTCGGCCT CAAAACAAGAT Anti-sense: ACTCCAGGAGGTG GAACAGTAGTCT |
| Sequence-based reagent | ds-*Boule* RNAi primer set for transcription template | *Amukamara et al., 2020* | PCR primers | Sense: AGCCT CACCACCAGT ATTCG Anti-sense: AGGGTGCC TAGGATTGGACT |
| Sequence-based reagent | qRT-PCR primer set: *Dnmt1* | *Amukamara et al., 2020* | PCR primers | Sense: GCTTGGA CAAAGGCTACTACT Anti-sense: CTTCGTGGTC CCTTATCCTTATC |
| Sequence-based reagent | qRT-PCR primer set: *Boule* | *Amukamara et al., 2020* | PCR primers | Sense: TATTCGT ACCACCCTCTTCC Anti-sense: GACAATGGCT GGGTCATAAG |

*Continued on next page*

*Continued*

| Reagent type (species) or resource | Designation | Source or reference | Identifiers | Additional information |
|---|---|---|---|---|
| Sequence-based reagent | qRT-PCR primer set: *Vasa* | *Amukamara et al., 2020* | PCR primers | Sense: CTGTTGCT CCTCAGGTTATT Anti-sense: CATTAAGCCTT CCAGGAGTAG |
| Sequence-based reagent | qRT-PCR primer set: *actin* | *Amukamara et al., 2020* | PCR primers | Sense: CTGTCTCCCG AAAGAGAATATG Anti-sense: TCTGTATGGAT TGGAGGATCTA |
| Sequence-based reagent | qRT-PCR primer set: *GAPDH* | *Amukamara et al., 2020* | PCR primers | Sense: ACGGTTTCAA GGAGAAGTTAG Anti-sense: AGCTGATGGTG CAGTTATG |
| Software, algorithm | JMP Pro | SAS Institute | Version 14.1 RRID:SCR_014242 | |
| Other | DAPI stain | Invitrogen | D1306 | 0.5 µg/mL |

## Animal care

All experimental animals were from colonies of laboratory reared *O. fasciatus* (Carolina Biologicals, Burlington, NC) and were reared under standard rearing conditions of 12 hr:12 hr light/dark at 27° C. To collect animals of known age and social conditions, eggs were removed from the mass colonies and allowed to hatch in plastic storage containers containing ad libitum deionized water and organic, raw sunflower seeds. For the nymph injections, nymphs were pulled from mixed sex nymph colonies at the third instar or fifth instar. For adult injections, nymphs were separated by sex at the fourth instar and housed in single sex colonies. These were checked daily and newly emerged adults. All experimental animals were placed into individual petri dishes with food and water.

## Developmental expression

Given our hypothesis that *Dnmt1* is required for development of germ cells and meiosis we examined the expression levels of *Dnmt1* across testis development in males. Groups of nymphs were staged and sexed. Testes were dissected from fourth and fifth instar nymphs and flash frozen in liquid nitrogen and stored at −80° C. We also collected testes from virgin males on the day of adult emergence and after sexual maturation at 7 days post-adult emergence. We also collected head, gut, and muscle samples from the sexually mature adult male individuals for tissue-specific expression.

Total RNA was extracted using a Qiagen RNA easy kit with Qiazol (Qiagen, Venlo, The Netherlands) and complementary DNA (cDNA) synthesized from 500 ng RNA with aScript cDNA Super-Mix (Quanta Biosciences, Gaithersburg, MD). Quantitative real-time PCR (qRT-PCR) was used to determine expression levels of *Dnmt1* and two genes with known functions in spermatogenesis, *Boule* and *Vasa*. Primers are described in *Amukamara et al., 2020*. As *Amukamara et al., 2020*, actin and GAPDH were used as reference genes. We have previously validated these reference genes and they are accepted as robust reference genes in *O. fasciatus* (*Meinzer et al., 2019*). We used a Roche LightCycler 480 with the SYBR Green Master Mix (Roche Applied Science Indianapolis, IN). All samples were run with three technical replicates using 10 µL reactions. There were 10 biological replicates for each stage. Each biological replicate of second, third, fourth, and fifth instar nymphs consisted of pools of 10, 5, 4, and 3 individuals, respectively. Adult replicates consisted of individual animals. Sample size was based on past experience balanced by the cost of qRT-PCR. Primer efficiency calculations, genomic contamination testing, and endogenous control gene selection were performed as described in *Cunningham et al., 2015*. We used the ΔΔCT method to compare levels of gene expression across the samples (*Livak and Schmittgen, 2001*). Gene expression was

standardized per individual to account for different numbers of individuals within each group at each developmental stage. Differences in expression levels were analyzed using ANOVA in JMP Pro v14. If there was a significant overall effect, we compared means using Tukey–Kramer HSD.

## RNAi preparation

Double-stranded RNAs were prepared as described in *Amukamara et al., 2020*. Briefly, DNA templates were prepared by PCR using gene-specific primers (*Amukamara et al., 2020*). Sense and anti-sense RNA were transcribed together with an Ambion MEGAscript kit (ThermoFisher Sci, Waltham, MA) and allowed to anneal to form a 404 bp ds-*Dnmt1* RNA. The concentration of dsRNA was adjusted to 3 µg/µL in injection buffer (5 mM KCl, 0.1 mM NaH$_2$PO$_4$).

## Nymph injections, testis size, and morphology

To examine the effect of *Dnmt1* knockdown prior to or following the wave of meiosis initiated in the fourth instar stage, nymphs were injected with ds-*Dnmt1* or control ds-RED injections at either the third instar or fifth instar stage of development. For further information on controls and testing for potential off-target effects please see *Amukamara et al., 2020*. Nymphs were anaesthetized at 4°C for 20 min prior to injection. Nymphs were injected in the abdomen using pulled glass capillary needles (Sutter Instrument Company micropipette puller model P-97, Novato, CA) between the third and fourth abdominal segments (*Chesebro et al., 2009*). Nymphs were injected with 2 µL volume for all injections. Following injections, nymphs were placed in individual petri dishes and monitored for development. Date of adult emergence was recorded. We did not do a power analysis, but based on preliminary data on the strength of the effect we aimed for 25 individual males for each treatment. Males were randomly assigned to a treatment group. Not all males survived to age of dissection, resulting in final sample sizes of 22 and 19 for third instar males injected with buffer and ds-*Dnmt1*, respectively and 25 and 20 for fifth instar males injected with buffer and ds-*Dnmt1*, respectively.

### Testis size

At 7–10 days post-adult emergence, virgin males were dissected and their testes were removed. Whole testes were allowed to settle into 1 mL phosphate buffered saline (PBS) and were imaged with a Leica M60 Stereomicroscope with Leica Application Suite software (LAS v4). Testis area was measured on one of the pair from each male with the LAS by outlining the whole testis with all seven testis tubules. Differences in testis area were analyzed using ANOVA in JMP Pro v14.

### Testis tubule structure

*O. fasciatus* testes contain seven individual testis tubules surrounded by a relatively impermeable, autofluorescent membrane. Individual testis tubules were removed from the outer membranous sheath for fixation and staining. Males from each treatment were dissected across four dissection days. Testis tubules from individual males within a treatment were pooled for staining. Thus each day one tube was a replicate with tubules from several individual males. Tubules were fixed for 30 min in 4% formaldehyde in PBS plus 0.1% Triton-X100 (PBT) and stained for evidence of cell division using an α-phosphohistone H3 Ser10 (pHH3) primary antibody (Millipore antibody 06–570, Sigma-Aldrich, St. Louis, MO). α-phosphohistone H3 (pHH3) stains for chromosome condensation in preparation for mitosis and meiosis (*Hans and Dimitrov, 2001*; *Prigent and Dimitrov, 2003*). The secondary antibody was an Alexa Fluor goat-anti-rabbit 647 (ThermoFisher Scientific, Waltham, MA). Following antibody staining the tubules were stained with DAPI (0.5 µg/mL PBT) to visualize nucleic acids. Stained tubules were mounted in Mowiol 4–88 mounting medium (Sigma-Aldrich, St. Louis, MO) and visualized with an Olympus BX51 Fluorescent microscope. Images were taken of every testis tubules present on each slide. Representative images are presented in the figures.

## Quantitative real time PCR

While we had evidence that all our RNAi treatments successfully knocked down expression in females (*Amukamara et al., 2020*), to confirm that our RNAi treatment was effective in males, total RNA and genomic DNA were extracted from flash frozen testes of a subsample of males 6–10 days

following injection with the dsRNA using a Qiagen Allprep DNA/RNA Mini Kit (Qiagen, Venlo, The Netherlands). Expression levels for *Dnmt1* were analyzed using qRT-PCR as described above.

## Quantification of DNA methylation

The DNA from each prep used for qRT-PCR was used to prepare MethylC-seq libraries as described in *Urich et al., 2015* and *Amukamara et al., 2020*. Samples were sequenced on a NextSeq500 and qualified reads were aligned to the *O. fasciatus* genome assembly according to previously published methods (*Schmitz et al., 2013*). The percent DNA methylation was calculated by dividing the total number of methylated CpG sites by the total number of CpG sites (*Schultz et al., 2012*). Spiked in Lambda DNA, which is fully demethylated, was used as a control for the sodium bisulfite conversion rate of unmodified cytosines. Three individuals were sequenced for each treatment.

## Adult injections

### RNAi treatment

Sexually mature virgin males (7 days post-adult emergence) were injected with 3 µL ds-*Dnmt1* RNA or control injections using a pulled glass capillary needle between the third and fourth abdominal segments (*Chesebro et al., 2009*). Control injections for the male fecundity experiments were injection buffer alone. We used ds-RED control injections for the testis imaging experiment. Previous studies from our lab have shown no difference in buffer alone controls or non-specific ds-RNA (*Bewick et al., 2019*). Following injection males were placed into individual petri dishes and provided with ad libitum food and water. We did not do a power analysis, but based on preliminary data on the strength of the effect we aimed for 30 individual males for each treatment. Males were randomly assigned to a treatment group.

### Male fecundity

Preliminary experiments had shown that two previous matings were required to deplete sperm stores. Therefore, in order to allow males to deplete sperm stores acquired during nymphal development and sexual maturation, males were placed in mating trials with three 7- to 10-day-old virgin females, one provided each week. The first female was placed in the male's petri on the day of injection with cotton wool as an oviposition substrate. The female and all eggs were removed at the end of the week (7 days post-injection) and discarded and replaced with a second 7- to 10-day-old virgin female and fresh cotton wool. Again, at the end of the week (14 days post-injection), the female and all resulting eggs were discarded. A third, focal, 7- to 10-day-old virgin female was placed with the male with fresh cotton wool. The third female and experimental male were given 1 week to mate before the male was removed from the petri dish for analysis of testis size and structure. The female was maintained for her lifetime, provided with ad libitum food and water. The eggs produced by the third female were collected from the petri dish twice a week at 3–4-day intervals and the oviposition substrate replaced with fresh cotton wool. The eggs collected were placed in a separate container and allowed to develop to hatching (approximately 7–10 days following collection) and then frozen at −20° C until assayed. We recorded the first collection date for which no eggs hatched. We analyzed the time to end of fertilized eggs due to sperm depletion relative to treatment using a survival analysis (Wilcoxon Rank Sum test) using JMP Pro v14.1. Not all males survived to date of dissection, or their mates died during the course of egg collection, requiring these males to be removed from the analysis, resulting in a sample size of 21 control and 20 ds-*Dnmt1*-treated males in the final fecundity analysis.

### Testis size

At the end of the 1-week mating trial with the third female (21 days post-injection), males were dissected and their testes removed into 1 mL PBS. Whole testes were photographed and measured as described above. Not all males survived to date of dissection, resulting in a final sample size of 28 control and 25 ds-*Dnmt1*-treated males.

### Testis tubule structure

Testis tubules from mated males were isolated, fixed, and stained as described above.

## Acknowledgements

The authors acknowledge Allen J Moore and other members of the Moore lab for many helpful discussions, and Luvika Gupta, whose undergraduate research provided preliminary data used in the experimental design. The authors would also like to acknowledge Tyler Earp for preparing and analyzing low throughput whole genome bisulfite sequencing data.

## Additional information

### Funding

No external funding was received for this work.

### Author contributions

Joshua T Washington, Katelyn R Cavender, Conceptualization, Data curation, Formal analysis, Investigation, Writing - original draft, Writing - review and editing; Ashley U Amukamara, Conceptualization, Data curation, Formal analysis, Investigation; Elizabeth C McKinney, Conceptualization, Data curation, Formal analysis, Investigation, Writing - review and editing; Robert J Schmitz, Conceptualization, Data curation, Formal analysis, Writing - review and editing; Patricia J Moore, Conceptualization, Data curation, Formal analysis, Supervision, Investigation, Writing - original draft, Writing - review and editing

### Author ORCIDs

Robert J Schmitz (iD) http://orcid.org/0000-0001-7538-6663
Patricia J Moore (iD) https://orcid.org/0000-0001-9802-7217

### Decision letter and Author response

Decision letter https://doi.org/10.7554/eLife.62202.sa1
Author response https://doi.org/10.7554/eLife.62202.sa2

## Additional files

### Supplementary files

• Transparent reporting form

### Data availability

Source data files have been provided for Figures 2, 3, 4, and 6.

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
