## [Decision Letter]

**Acceptance summary:**

This very thorough paper examines the role of Dmnt1 in Oncopeltus, an emerging insect model. The questions addressed are interesting and important, and the authors have developed a good system including Dmnt1 knockdowns, with which to examine these questions.

**Decision letter after peer review:**

Thank you for submitting your article "The essential role of Dnmt1 in gametogenesis in the large milkweed bug Oncopeltus fasciatus" for consideration by *eLife*. Your article has been reviewed by 2 peer reviewers, and the evaluation has been overseen by Michael Eisen as the Senior and Reviewing Editor. The reviewers have opted to remain anonymous.

The reviewers have discussed the reviews with one another and the Reviewing Editor has drafted this decision to help you prepare a revised submission.

This paper examines the role of Dmnt1 in Oncopeltus, an emerging insect model. The manuscript is very thorough and well written. The analyses are generally well performed. The questions here are interesting and important, and the authors have developed a good system in which to examine these questions. This includes effective RNAi knockdown to probe Dmnt1 function.

However, it is our judgment that significant additional experiments are required to support the claims, and that the findings are overly generalized to other insects without sufficient data to support such claims. If you choose to submit a revision, it must successfully address the points raised below.

Essential revisions

1. The authors predict that "if Dnmt1 is required during gametogenesis, particularly meiosis, then its expression should mirror those of two genes known to be involved in germ cell development, Boule and Vasa." This may or may not be the case, but either way, the outcome has no predictive value as to function because expression was examined in whole animals. The expression of genes that are tissue-specific might be expected to follow the profile of other genes specific to that tissue, but genes expressed in multiple tissues could show any expression profile in whole animals depending on timing and level of expression in each tissue.

Therefore, Figure 1 does not speak clearly to the claims paper. Rather, tissue-specific expression (RT-PCR and/or in situ) would be required to show that Dmnt1 is expressed in the male germline. Similarly, "Expression of Dnmt1 in whole body samples was lower in sexually mature adult males than sexually mature adult females, but male and female nymphs had the same 84 expression levels" is not informative, since many aspects of physiology and development vary between adult males and females (e.g., many genes expressed in the brain are sexually dimporphic).

2. The authors appear to have effectively used RNAi to knock down Dmnt1 in the testis (Figure 2) but it is generally accepted that a control using an unrelated dsRNA should be done for such experiment. The methods section statement that previous studies showed no difference between buffer and non-specific dsRNA as controls is not relevant here because those experiments did not test the same parameters (i.e., testis) as this current study.

3. RNAi knockdown in 3rd nymphal stage but not fifth resulted in reduced testis size. However, the injection per se reduced testis size – I assume likely due to the site at which the needle is injected. The effect of the specific dsRNA does show statistical significance but the experimental design is problematic in that the injection itself is causing the same phenotype as is being scored for the dsRNA effect.

4. The data presented in Figure 4 and 6 are intriguing. However the images are difficult to interpret for those not familiar with looking at Oncopeltus testis. The PI's website has an image that is much clearer – one can view individual cells, stained with the same markers – than Figure 4 and 6. We suggest that the imaging be improved here and I suspect they will more strongly support the conclusions made. It would also be helpful to include schematic drawings of the structures/cell types to help guide the viewers.

5. In the discussion: It is an overstatement to say things like "without any obvious phenotypic effects in somatic tissue" and nymphs "develop normally and are indistinguishable from control adult." No tissue other than the testis were described in enough detail to say there were no other effects. If such observations were done, they should be reported here.

6. The claim that "Our results raise the question of how a gene so critical in fitness across multiple insect species can have diverged widely across the insect tree of life."

This question has been raised already before as the authors acknowledge by citing some of the key studies (e.g. Lyko 2018, Bewick et al. 2019, Glastad et al. 2019). However, that the gene is critical for fitness is only shown for one species, so this is too generalized a claim.

Additional comments/questions:

1. Introduction, page 4: "Testing the function of Dnmt1 in males allows us to capitalize on the well-characteristized process of spermatogenesis"

I am far of an expert on spermatogenesis and to me this process is not entirely clear. I suggest to give a reference if the authors mean O. fasciatus or insects in general and to make clear if the study really allows claims about 'insects in general'.

2. page 4: "Our results demonstrated that Dnmt1 is required…". I wondered at this point if the gene expression of Dnmt1 is meant or if there was also a study of impact on DNA methylation. Please clarify.

3. Discussion, page 12, line 206: "number of insect groups the maintenance methyltransferase DNMT1 is essential to spermatogenesis". Please reference which insect groups and give a reference.

4. Discussion, page 15, line 268: "However, we do not think that the function of Dnmt1 in spermatogenesis is unique to *O. fasciatus*." What is the reasoning for this claim? What is the evidence besides personal opinion?

5. Conclusion, lines 325-327. I certainly agree that functional analysis on different functions for Dnmt1 would be relevant, the question itself is not new (e.g. Lyko 2018, Provataris et al. 2018).

6. Page 13, 221: "..spermatogenesis should be affected by Dnmt1 expression as well as oogenesis." Do I understand correctly that in this study there was tested for spermatogenesis only? Please clarify and write clearly.

7. Page 14, line 252: "The phenotype of knockdown of Dnmt1 is exacerbated when knockdown occurs prior to the stage of testis development when meiosis occurs, as was observed in female *O. fasciatus*." I assume in female O. fasciatus there was also an exacerbation of phenotype of knockdown, but it reads as there was also worse when knockdown occurred prior to the stage of testis development in females – which does not make sense.

---

## [Author Response]

Essential revisions1. The authors predict that "if Dnmt1 is required during gametogenesis, particularly meiosis, then its expression should mirror those of two genes known to be involved in germ cell development, Boule and Vasa." This may or may not be the case, but either way, the outcome has no predictive value as to function because expression was examined in whole animals. The expression of genes that are tissue-specific might be expected to follow the profile of other genes specific to that tissue, but genes expressed in multiple tissues could show any expression profile in whole animals depending on timing and level of expression in each tissue.Therefore, Figure 1 does not speak clearly to the claims paper. Rather, tissue-specific expression (RT-PCR and/or in situ) would be required to show that Dmnt1 is expressed in the male germline.

We agree with this comment and thank the reviewer with this suggestion. In the revision we have examined testis-specific expression of *Dnmt1*, as suggested. The new data show that the tissue specific expression matches our prediction of similar expression patterns to other genes involved in germ cell development. We have provided a revised Figure 1 (which is now Figure 2) documenting testis-specific expression across the key developmental stages for *Dnmt1*, *Boule*, and *Vasa*. We have also revised the manuscript text to reflect the change in methods (Lines 392-394) and results (lines 70-80).

Similarly, "Expression of Dnmt1 in whole body samples was lower in sexually mature adult males than sexually mature adult females, but male and female nymphs had the same 84 expression levels" is not informative, since many aspects of physiology and development vary between adult males and females (e.g., many genes expressed in the brain are sexually dimporphic).

The reviewer raises a valid point. To address this comment, we have revised the manuscript to include tissue specific expression data from males showing that *Dnmt1*, while expressed across all body tissues, has its highest expression in the testes, as has been shown for females in which the expression of *Dnmt1* is highest in the ovaries. We have included these data in the main body of the manuscript (Lines 77-80) and also updated the supplementary figure to reflect this new data. We have data that shows that ovary expression is higher than testis expression, but agree with the reviewer that this comparison isn’t informative and wasn’t required for the manuscript. So we have removed the comparison between males and females.

2. The authors appear to have effectively used RNAi to knock down Dmnt1 in the testis (Figure 2) but it is generally accepted that a control using an unrelated dsRNA should be done for such experiment. The methods section statement that previous studies showed no difference between buffer and non-specific dsRNA as controls is not relevant here because those experiments did not test the same parameters (i.e., testis) as this current study.

We have repeated a number of the experiments using a non-specific dsRNA control. There was no significant difference in buffer injected and dsRED injected treatments. We have shown that *Dnmt1* is knocked down in ds*Dnmt1* treated males relative to dsRED treated males (Figure 3). In the revision we are clear where we used dsRED and the one experiment (male fecundity, Figure 6) where we were unable to redo the experiment due to limitations in time and personnel.

3. RNAi knockdown in 3rd nymphal stage but not fifth resulted in reduced testis size. However, the injection per se reduced testis size – I assume likely due to the site at which the needle is injected. The effect of the specific dsRNA does show statistical significance but the experimental design is problematic in that the injection itself is causing the same phenotype as is being scored for the dsRNA effect.

We appreciate the reviewer for raising this concern. However, in repeating this experiment with dsRED, we also included uninjected and buffer injected controls. We saw no effect on testis size in any treatment other than the ds*Dnmt1* knockdown. We are confident that this reflects our increased expertise in injecting the 3^rd^ instar nymphs and change in injection site over the past year. We have revised Figure 3 (now Figure 4) and the text (lines 142-143) to reflect this result. In addition, we have included the uninjected control on the graph and in the analysis to document that the dsRED treatment has no effect on the phenotype under study.

4. The data presented in Figure 4 and 6 are intriguing. However the images are difficult to interpret for those not familiar with looking at Oncopeltus testis. The PI's website has an image that is much clearer – one can view individual cells, stained with the same markers – than Figure 4 and 6. We suggest that the imaging be improved here and I suspect they will more strongly support the conclusions made. It would also be helpful to include schematic drawings of the structures/cell types to help guide the viewers.

This is an excellent point. We have included an introductory figure (now Figure 1) that shows both a simple confocal image of an *Oncopeltus* testis tubule alongside a schematic drawing showing the regions of interest within the testis tubule. We hope that this will help guide the readers through the developmental processes that we are discussing in the manuscript.

We apologize for the initial images not being to our normal high standard. Unfortunately, while we were preparing the manuscript the confocal facility was closed due to the pandemic and we were not sure when it would be available. With the reopening of the facility we were able to collect confocal images on the ds*Dnmt1* treated testes and dsRED control testes.

5. In the discussion: It is an overstatement to say things like "without any obvious phenotypic effects in somatic tissue" and nymphs "develop normally and are indistinguishable from control adult." No tissue other than the testis were described in enough detail to say there were no other effects. If such observations were done, they should be reported here.

We have removed both these statements (Lines 241-242).

6. The claim that "Our results raise the question of how a gene so critical in fitness across multiple insect species can have diverged widely across the insect tree of life."This question has been raised already before as the authors acknowledge by citing some of the key studies (e.g. Lyko 2018, Bewick et al. 2019, Glastad et al. 2019). However, that the gene is critical for fitness is only shown for one species, so this is too generalized a claim.

We acknowledge that this claim has only been functionally tested in two species (Line 235 and 321) and have bolstered our reasoning for suspecting that *Dnmt1* may have a function in spermatogenesis and gametogenesis more generally through our response to other comments (Additional Comment 3, Lines 235-240; Additional Comment 4, Lines 308-310).

Additional comments/questions:1. Introduction, page 4: "Testing the function of Dnmt1 in males allows us to capitalize on the well-characteristized process of spermatogenesis"I am far of an expert on spermatogenesis and to me this process is not entirely clear. I suggest to give a reference if the authors mean *O. fasciatus* or insects in general and to make clear if the study really allows claims about 'insects in general'.

We have clarified that we are drawing on both the specific information from *O. fasciatus* and general conservation of spermatogenesis in the insects. We have added a reference (lines 3840).

2. page 4: "Our results demonstrated that Dnmt1 is required…". I wondered at this point if the gene expression of Dnmt1 is meant or if there was also a study of impact on DNA methylation. Please clarify.

We have clarified that we are examining the requirement for the expression of *Dnmt1* (line 57), although we have shown that reduced expression of *Dnmt1* results in reduced methylation, so these two factors are cofounded, as discussed.

3. Discussion, page 12, line 206: "number of insect groups the maintenance methyltransferase DNMT1 is essential to spermatogenesis". Please reference which insect groups and give a reference.

This sentence has been rephrased to acknowledge while functional studies have only been done in two species, expression studies have indicated a role for *Dnmt1* in reproduction in other species. We have added the species and citations (Lines 235-240).

4. Discussion, page 15, line 268: "However, we do not think that the function of Dnmt1 in spermatogenesis is unique to *O. fasciatus*." What is the reasoning for this claim? What is the evidence besides personal opinion?

We have added citations that indicate testis-elevated expression of Dnmt1 in other insects to clarify our reasoning for this statement (Lines 308-310).

5. Conclusion, lines 325-327. I certainly agree that functional analysis on different functions for Dnmt1 would be relevant, the question itself is not new (e.g. Lyko 2018, Provataris et al. 2018).

We have added these citations.

6. Page 13, 221: "..spermatogenesis should be affected by Dnmt1 expression as well as oogenesis." Do I understand correctly that in this study there was tested for spermatogenesis only? Please clarify and write clearly.

We have clarified this statement (Lines 259-260).

7. Page 14, line 252: "The phenotype of knockdown of Dnmt1 is exacerbated when knockdown occurs prior to the stage of testis development when meiosis occurs, as was observed in female *O. fasciatus*." I assume in female *O. fasciatus* there was also an exacerbation of phenotype of knockdown, but it reads as there was also worse when knockdown occurred prior to the stage of testis development in females – which does not make sense.

We have revised this statement for clarity (Lines 290-292).